# Predominance of *BRCA2* Mutation and Estrogen Receptor Positivity in Unselected Breast Cancer with *BRCA1* or *BRCA2* Mutation

**DOI:** 10.3390/cancers14133266

**Published:** 2022-07-04

**Authors:** Pascal Pujol, Kevin Yauy, Amandine Coffy, Nicolas Duforet-Frebourg, Sana Gabteni, Jean-Pierre Daurès, Frédérique Penault Llorca, Frédéric Thomas, Kevin Hughes, Clare Turnbull, Virginie Galibert, Chloé Rideau, Carole Corsini, Laetitia Collet, Benoit You, David Geneviève, Nicolas Philippe

**Affiliations:** 1Department of Cancer Genetics, CHU Montpellier, Université de Montpellier, 34000 Montpellier, France; gabtenisana1992@gmail.com (S.G.); v-galibert@chu-montpellier.fr (V.G.); c-rideau@chu-montpellier.fr (C.R.); c-corsini@chu-montpellier.fr (C.C.); 2CREEC, UMR IRD 224-CNRS 5290 Université Montpellier, 34000 Montpellier, France; frederic.thomas2@ird.fr; 3SeqOne Genomics, 34000 Montpellier, France; kevin.yauy@seqone.com (K.Y.); nicolas.duforet@seqone.com (N.D.-F.); nicolas.philippe@seqone.com (N.P.); 4Centre de recherche UGA, Institute of Advanced Biosciences, Inserm U 1209, CNRS UMR 5309, 38000 Grenoble, France; 5Unité de Recherche Clinique LMNT-AESIO Santé, 34000 Montpellier, France; a.coffy@languedoc-mutualite.fr (A.C.); jp.daures@languedoc-mutualite.fr (J.-P.D.); 6INSERM U1240, Centre Jean Perrin, Department of BioPathology, University Clermont Auvergne, 63000 Clermont-Ferrand, France; frederique.penault-llorca@clermont.unicancer.fr; 7Division of Surgical Oncology, Massachusetts General Hospital and Harvard Medical School, Boston, MA 02114, USA; kshughes@partners.org; 8 Institute of Cancer Research, 15 Cotswold Rd, London SM2 5NG, UK; clare.turnbull@icr.ac.uk; 9Institut de Cancérologie des Hospices Civils de Lyon (IC-HCL), Faculté de Médecine Lyon-Sud, Université Claude Bernard Lyon 1, 69002 Lyon, France; laetitia.collet@lyon.unicancer.fr (L.C.); benoit.you@chu-lyon.fr (B.Y.); 10Département de Génétique Médicale, Maladies Rares et Médecine Personnalisée, CHU Montpellier, Université de Montpellier, 34000 Montpellier, France; d-genevieve@chu-montpellier.fr; 11INSERM U1183, Université de Montpellier, 34000 Montpellier, France

**Keywords:** *BRCA1*, *BRCA2*, breast cancer, estrogen receptor

## Abstract

**Simple Summary:**

We performed a meataanalysis of BRCA1 or BRCA2 germline pathogenic or likely pathogenic variant (gBRCA) in 108,699 unselected breast cancer patients and in 238,972 unaffected individuals. The metanalysis shows that 3.4% unselected breast cancer patients have a gBRCA. In unselected breast cancer patients with gBRCA, more than half of tumors are estrogen receptor-positive. 0.5% of unaffected individuals of the studied populations are gBRCA carriers. The frequency of gBRCA2 and gBRCA1 heterozygosity is estimated at 1/288 and 1/434, respectively. In unselected breast cancer and in unfafected individuals gBRCA2 is more frequent than gBRCA1.

**Abstract:**

Background: Poly(ADP-ribose) polymerase 1 inhibitor (PARPi) agents can improve progression-free survival of patients with breast cancer who carry a germline BRCA1 or BRCA2 pathogenic or likely pathogenic variant (gBRCA) in both the metastatic and adjuvant setting. Therefore, we need to reassess the frequency of gBRCA1 and gBRCA2 in order to redefine the criteria for women and tumor phenotype that should be tested. Objective: We studied the relative distribution of gBRCA1 and gBRCA2 in unselected populations of women with breast cancer and in unaffected individuals. We also analyzed the proportion of estrogen receptor (ER)-positive (ER+) tumors in unselected breast cancer patients with gBRCA. Design: We performed a meta-analysis of studies of unselected breast cancer that analyzed the relative contribution of gBRCA1 versus gBRCA2 among unselected breast cancer cases in gBRCA carriers. We then performed a meta-analysis of gBRCA carriage in unaffected individuals from genome-wide population studies, the gnomAD databank, and case–control studies. Results: The *BRCA2* gene was involved in 54% of breast cancer cases in unselected patients with gBRCA (n = 108,699) and 60% of unaffected individuals (n = 238,973) as compared with 38% of the largest gBRCA family cohort (n = 29,700). The meta-analysis showed that 1.66% (95% CI 1.08–2.54) and 1.71% (95% CI 1.33–2.2) of unselected breast cancer patients carried gBRCA1 and gBRCA2, respectively. In a population of unaffected individuals, the frequency of heterozygosity for gBRCA1 and gBRCA2 was estimated at 1/434 and 1/288, respectively. Nearly 0.5% of unaffected individuals in the studied populations carried a gBRCA. Carriage of a gBRCA was 2.5% for patients with ER+ tumors (95% CI 1.5–4.1) and 5.7% (95% CI 5.1–6.2) for those with ER- tumors. Overall, 58% of breast tumors occurring in women carrying a gBRCA were ER+ (n = 86,870). Conclusions: This meta-analysis showed that gBRCA2 carriage is predominant in unselected breast cancer patients and unaffected individuals. ER+ tumors among women with gBRCA-related breast cancer are predominant and have been underestimated. Because PARPi agents improve progression-free survival with ER+ gBRCA breast cancer in most clinical trials, breast cancer should be considered, regardless of ER status, for *BRCA1/2* screening for therapeutic purposes.

## 1. Introduction

An estimated 10% of breast cancer cases likely result from hereditary causes [1]. *BRCA1* and *BRCA2* account for the most important genes identified for risk of the disease [2,3,4,5]. However, it is likely that less than 10% of *BRCA1* or *BRCA2* mutation carriers have been identified, and genetic testing criteria could lead to an underdiagnosis of hereditary breast and ovarian cancer [6,7,8,9].

Poly (ADP-ribose) polymerase inhibitor (PARPi) agents have been found effective for treating high-risk and metastatic breast cancer related to the germline *BRCA1* or *BRCA2* pathogenic or likely pathogenic variant (gBRCA, [10,11,12,13]). Recently, the results of the OlympiA randomized phase III trial showed that adjuvant olaparib significantly improved overall survival as well as invasive disease-free survival and distant disease-free survival in patients with gBRCA and high-risk HER2-negative early breast cancer [14]. Therefore, we need to identify which breast cancer patient and which tumor type could benefit from genetic screening. 

In 70% of cases, breast cancer related to the germline *BRCA1* pathogenic or likely pathogenic variant (gBRCA1) has an estrogen receptor-negative (ER-) phenotype [15]. In the largest available family cohorts, among families identified with gBRCA, gBRCA1 is the predominant molecular form, and is implicated in 62% of 29,700 identified families [16]. Because of these features and other therapeutic options available for ER+ breast cancer, screening for gBRCA in ER+ breast cancer has not been prioritized and most attention has been paid to genetic testing of ER- tumors for theragnostic purposes.

However, in most subgroup analyses of gBRCA patients with metastatic hormone-resistant ER+ breast cancer, PARPi agents have increased both progression-free survival and quality of life as compared with chemotherapy ([11,12,13,17]; Appendix A). In the Embraca and Brocade studies, the risk of progression in patients with ER+/HER2- metastatic breast cancer was decreased by 33% (hazard ratio (HR) 0.67, 95% confidence interval (CI) 0.32–0.71) and 31% (HR 0.69, 95% CI 0.52–0.92), respectively [11,13]. In the subgroup analysis of the OlympiAD trial, olaparib was not associated with improved progression-free survival in patients with ER+/HER2- metastatic breast cancer (HR 0.82, 95% CI 0.55–1.26), despite a 65.4% overall response rate [12]. Of note, a high proportion of patients had already received chemotherapy (77%). 

In the adjuvant setting, progression-free survival was improved but not significantly in high-risk ER+ breast cancer patients who received olaparib and no neoadjuvant chemotherapy (HR 0.52, 95% CI 0.25–1.04) [10]. Therefore, the question of using PARPi agents in ER+ gBRCA breast cancer is of major clinical relevance.

In the literature on BRCA testing in breast cancer series, therapeutic trials or from accrual by oncogenetic consultation, a bias in recruitment could not be ruled out. It is likely that more testing is performed in patients or individuals with a family history of ovarian cancer, triple negative breast tumors or young age. As compared with gBRCA2 carriage, gBRCA1 carriage is associated with increased risk of ovarian cancer, young age at breast cancer occurrence and increased proportion of ER- breast tumors, which may affect the relative distribution of gBRCA1 or gBRCA2 as well as the frequency of ER+ breast cancer. 

We performed a meta-analysis of available cohorts of unselected breast cancer in the literature addressing the relative contribution of gBRCA1 and gBRCA2 and ER+ tumors in gBRCA patients overall. We then evaluated the prevalence of gBRCA carriage in unaffected individuals by a meta-analysis of genome-wide population studies, the gnomAD sequencing aggregation database and control groups of breast cancer case–control studies.

## 2. Patients and Methods

### 2.1. Literature Search Strategy

The PubMed database was searched for English-language studies from January 2000 to August 2021 by using the following query of terms related to gBRCA and ER status frequency in patients with breast cancer: ((gene, BRCA 1[MeSH]) OR (gene, BRCA 2[MeSH]) OR (BRCA 1 gene[MeSH]) OR (BRCA 2[MeSH]) OR (BRCA1 2[MeSH]) OR (breast cancer 1 gene[Tw]) OR (breast cancer 2 gene[tw]) OR (*BRCA1/2*[MeSH])) AND ((breast cancerg[MeSH]) OR (breast tumor[MeSH]) OR (estrogen receptor[MeSH]) OR (case control[MeSH]) OR (prevalence[MeSH]) OR (unselected[Tw]) OR (general population[Tw]) OR (unaffected[Tw]) AND ((pathogenic variant[MeSH]) OR (likely pathogenic variant[MeSH]) OR (germline mutation [MeSH]) OR (tumor mutation[Tw]) OR (somatic mutation[Tw])) AND (english[Language]) AND (“1995/01/01”[Date—Publication]: “2020/06/20”[Date—Publication]) NOT (case reports[Publication Type]) NOT (case reports[Tiab]) NOT (mice[Tw]). The literature search used variations and Boolean connectors of key terms. The results of the database searches were supplemented by searching bibliographies of seminal articles or reviews and contributions from expert panel members’ references.

We retained only unselected breast cancer studies analyzing gBRCA1 or gBRCA2. To avoid potential selection bias, we excluded studies of women with breast cancer who were undergoing genetic testing with specific criteria such as family history, young age and screening for therapeutic trials with potential enrichment for triple-negative breast cancer, studies of metastatic cancer, and studies of breast cancer in specific ethnicities (e.g., French Canadian or Ashkenazi Jewish heritage). Studies of somatic mutation were retained only if gBRCA data were available. The Quoron flowchart is given in Appendix A. A total of 2669 results were found. From these, 240 records were selected for full-text review and 31 publications were retained (cf. Appendix A). 

### 2.2. Meta-Analyses

We used Comprehensive Meta-Analysis software to estimate the pooled values of the parameters as well as their 95% confidence intervals (CIs). We estimated heterogeneity by using the Cochran Q test with the point estimate I^2^. I^2^ is the proportion of total variation contributed by between-study variation. If heterogeneity was present (Q statistic significant at 5%), we took it into account by using a random-effects model. Publication bias was visually estimated with funnel plots and quantified with the Egger test and the “trim-and-fill” method of Duval and Tweedie. Details of the meta-analyses are given in Appendix A. We performed subgroup analyses and used meta-regression models to evaluate potential sources of heterogeneity. We also conducted analyses stratified by ER status. Summary relative risks with 95% CIs were calculated with the method of DerSimonian and Laird by use of the assumptions of a random-effects model, which considers both within- and between-study variation. 

For the meta-analysis of gBRCA in unselected breast cancer, the following studies were retained [2,4,5,18,19,20,21,22,23,24]. For the meta-analysis of ER+/ER- tumors among unselected breast cancer patients, the following studies were retained [2,4,5,18,24,25].

Carriage of gBRCA was explored in the exome studies [26,27], in control cases of case–control studies [4,5] and in the exome cohort (n = 125,747) of gnomAD v2.1.1 [22]. gBRCA1 and gBRCA2 were identified in the ClinVar database ([28], August 2021 release). 

Only likely pathogenic and pathogenic variants from the ClinVar database were considered carrier variants. This decision was made because of premature truncating codons in gnomAD classified as variants of uncertain significance, conflicting, likely benign, or benign in ClinVar, and their prevalence is high (351 allele counts with 12 different premature truncating codons variants in BRCA1, 2048 allele counts with 11 different premature truncating codons variants in BRCA2). Premature truncating codons variants not available in ClinVar were not retained because their American College of Medical Genetics and Genomics classification is still uncertain and their prevalence in the database is low (one allele count with one variant in *BRCA1*, 16 allele counts with 14 variants in *BRCA2*).

## 3. Results

### 3.1. Carriage of gBRCA in Unselected Breast Cancer Patients

A total of 10 studies of unselected breast cancer series were retained on the basis of the selection criteria described in the Quoron flowchart (Appendix A). Meta-analysis of these studies of 108,699 unselected breast cancer cases showed that 3.4% of breast cancers occurred in patients with a germline pathogenic variant of *BRCA1* or *BRCA2* (95% CI 2.5–4.7, Figure 1a, Appendix A). There was no evidence of publication bias in the funnel plot (Appendix A, Meta-Analysis S8) or by Egger’s test (*p* = 0.23).

gBRCA1 and gBRCA2 status accounted for 1.66% (95% CI 1.08–2.54) and 1.71% (95% CI 1.33–2.2) of unselected breast cancer cases (Figure 1b,c, and Appendix A). There was no evidence of publication bias in the funnel plot (Appendix A, S9 and S10 for gBRCA1 and gBRCA2, respectively) or by Egger’s test (*p* = 0.92 and *p* =0.99 for gBRCA1 and gBRCA2, respectively).

Among breast cancer patients with gBRCA, 46% (n = 1496) carried a gBRCA1 and 54% (n = 1776) a gBRCA2 (Appendix A).

### 3.2. Frequency of ER+ Tumors among Unselected Breast Cancer Patients with gBRCA Mutation

The meta-analysis of 7 studies including a total of 86,870 patients showed that 2.5% of patients with ER+ tumors (95% CI 1.5–4.1, Figure 2a) carried a gBRCA1 or gBRCA2 pathogenic variant. There was no evidence of publication bias in the funnel plot (Appendix A, Meta-Analysis S6) or by Egger’s test (*p* = 0.46).

In total, 5.7% of patients with ER- tumors (95% CI 5.1–6.2, Figure 2b) carried a gBRCA1 or gBRCA2 pathogenic variant. There was no evidence of publication bias in the funnel plot (Appendix A, Meta-Analysis S7) or by Egger’s test (*p* = 0.43).

Among gBRCA1 carriers, 38% (457/1218) had ER+ tumors, whereas among gBRCA2 carriers, 75% (1085/1453) had ER+ tumors (Appendix A). Overall, 58% (1542/2671 of breast tumors with gBRCA carriage were ER+ (Appendix A) and 42% were ER-(1129/2671). Figure 2c represents the frequency of ER+ and ER- according to gBRCA1 or gBRCA2 carriage among unselected breast cancer patients. gBRCA1-ER+, gBRCA2-ER+, gBRCA1-ER-, gBRCA2-ER- status accounted for 17%, 41%, 28% and 14% of gBRCA breast cancers, respectively (Appendix A). 

### 3.3. Frequency of gBRCA Carriers among Unaffected Individuals 

We performed a meta-analysis of available data from genome-wide population studies [26,27], the gnomAD databank [28] and for unaffected individuals of case–control studies of genetic testing in breast cancer [4,15]. Only likely pathogenic and pathogenic variants from the ClinVar database were considered carrier variants.

Carriage of a gBRCA1 or BRCA2 pathogenic variant in the population of 238,972 individuals was 0.5% (95% CI 0.4–0.6) (Figure 3a, Appendix A). There was no evidence of publication bias in the funnel plot (Appendix A, Meta-Analysis S8) or by Egger’s test (*p* = 0.23).

The frequency of gBRCA1 and gBRCA2 was 0.18% (95% CI 0.12–0.25) and 0.32% (95% CI 0.27–0.38), respectively (Figure 3b,c, Appendix A, S3, S4). There was no evidence of publication bias in the funnel plot (Appendix A, Meta-Analysis S9 and S10, respectively) or by Egger’s test.

The frequency of heterozygous gBRCA1 and gBRCA2 was estimated at 1/434 and 1/288, respectively (Appendix A). The *BRCA2* gene accounted for 60% of all gBRCA carriers in the general population (Figure 2c, Appendix A). 

### 3.4. Comparison of gBRCA1 and gBRCA2 Contribution in Family Cohorts, Unselected Breast Cancer Patients and Unaffected Individuals

Figure 4 compares the relative distribution of gBRCA1 and gBRCA2 in the large family cohort from CIMBA ([16], n = 29,700), in unselected breast cancer patients (n = 86,870) and in unaffected individuals (n = 238,972). Although gBRCA1 accounted for 62% of identified families in the CIMBA cohort [16], gBRCA2 was the predominant form in unselected breast cancer patients and unaffected individuals (54% and 60%, respectively).

## 4. Discussion

Our overall finding of a higher gBRCA2/gBRCA1 ratio in the breast cancer population agrees with the most recent and largest studies of BRCA screening of unselected breast cancer [4,5]. Because most BRCA1 cases of breast cancer are ER- and most BRCA2 cases are ER+ [15], the frequency of ER+ tumors among the overall BRCA breast cancer patients depends on the relative contribution of the gBRCA2 incidence in unselected breast cancer patients. In several published reports describing the frequency of gBRCA in breast cancer series, the bias of selection due to family criteria but also age or triple-negative phenotype may have underestimated the frequency of gBRCA2. 

Our finding of a predominant prevalence of gBRCA2 versus gBRCA1 in the general population also agrees with recent unselected population-genomic screenings showing a higher-than-expected prevalence of gBRCA2 versus gBRCA1 in individuals of predominantly European ancestry [26,27]. Further studies in other populations are needed to detail the prevalence of gBRCA1 and gBRCA2 by ancestry.

In ER+/HER2- metastatic breast cancer, the use of CDK4/6 inhibitors has been associated with striking progress in survival and has become the standard first-line treatment combined with endocrine therapy, such as tamoxifen or fulvestrant in pre- or perimenopausal women or aromatase inhibitors in post-menopausal women [29]. However, 15% to 30% of patients do not respond to CDK4/6 inhibitors and exhibit disease progression within 24 weeks of treatment [29]. There is growing evidence that patients with gBRCA ER+/HER2- advanced breast cancer are at higher risk of early disease progression under CDK4/6 inhibitors than their counterparts with wild-type BRCA disease [30,31]. A recent retrospective study demonstrated significantly shorter overall survival for patients with gBRCA versus those with wild-type BRCA treated with CDK4/6 inhibitors combined with endocrine therapy (stratified HR 1.5, 95% CI 1.06–2.14) [30]. Time to therapy or death was also shorter although not significantly for these patients (stratified HR 1.24, 95% CI 0.96–1.59) [30]. In line with these results, a subgroup analysis of the PADA-1 trial showed that patients with BRCA-mutated disease treated with palbociclib and aromatase inhibitors experienced shorter median progression-free survival than those with wild-type BRCA disease: 14.3 versus 26.7 months [32]. The optimal sequence of endocrine-based therapy, platinum chemotherapy and PARPi agents in women with ER+ gBRCA advanced breast cancer is uncertain. However, given that CDK4/6 inhibitors combined with endocrine treatment improved both progression-free survival and overall survival and had a good toxicity profile, CDK4/6 inhibitors are considered the preferred option in ER+ gBRCA breast cancer. Further studies are needed to define the optimal place of PARP inhibition with regard to a CDKA/4 inhibitor and platinum regimen in the treatment of ER+ gBRCA advanced breast cancer. BRCA testing should be proposed to any patient with ER+ advanced breast cancer and resistance to endocrine therapy [33,34,35] because PARPi therapy represents a treatment option in gBRCA patients and has a higher benefit/risk ratio than chemotherapy, as recently recognized by ESMO [36,37]. 

It is likely that not all breast tumors display a double hit feature, which suggests that a subset of the BRCA-related breast cancer may develop (and possibly be sensitive to PARPi) in the absence of a BRCA second event [38,39]. Maxwell et al. [40] analyzed *BRCA* germline-mutated breast and ovarian tumors and found that although gBRCA1 breast and ovarian tumors had loss of heterozygosity in 90% and 93% of cases, respectively, gBRCA2 tumors retained the wild-type allele in 16% of all gBRCA2 ovarian tumors and 46% of gBRCA2 breast tumors. Thus, the question of a second hit in terms of ER phenotype deserves to be further investigated. 

In the adjuvant setting, the positive outcomes of the OlympiA trial in terms of progression-free survival but also overall survival [14] suggest that PARPi therapy is becoming a standard adjuvant treatment in patients with high-risk localized ER+/HER2- cancers. The eligibility criteria of this trial for patients with ER+ tumor were (1) if neoadjuvant chemotherapy, incomplete response (PCR) and CPS + EG stage >2 and (2) if no neoadjuvant therapy, >3 positive nodes (pN > 1). These findings led to a recent update of the American Society of Clinical Oncology guidelines [35,37]. Additional studies on the benefit of PARPi therapy in the adjuvant setting of gBRCA patients are required, including exploration of the potential benefit of PARPi therapy in lower-risk ER+ patients. 

Because of the clinical benefits of PARPi therapy in BRCA-related cancer, the lack of timely identification of a BRCA mutation represents a lost opportunity for patients. From 30% to 50% of patients with breast but also ovarian, prostate, or pancreatic cancer do not fulfill personal or family criteria for usual preventive BRCA testing [33]; thus, family history or personal criteria including triple negativity and young age at breast cancer diagnosis cannot be retained to select patients who require testing.

The higher incidence of gBRCA1 found in the largest international cohort of identified gBRCA carriers [16] could be due to lack of penetrance of breast cancer with gBRCA2 carriage. However, the prospective study by Kuckenbacher et al. [41] did not support this hypothesis for breast cancer incidence because the cumulative lifetime risk is similar (cumulative breast cancer risk to age 80 years: 72% and 69% for gBRCA1 and gBRCA2 mutations carriers, respectively). More likely, because the occurrence of breast cancer in gBRCA2 carriers is time-delayed as compared with gBRCA1 carriers, the older age of women at diagnosis could result in a relative lack of referral of testing for gBRCA2 as compared with gBRCA1. Another potential explanation for a higher testing rate in gBRCA1 is that the higher incidence of ovarian cancer with gBRCA1 versus gBRCA2 carriage (cumulative risk to age 80 years is 44% and 17%, respectively) could lead to more active BRCA testing in women with the criterion of a close relative with ovarian cancer. Finally, the well-known triple-negative phenotype of BRCA1 tumors could also result in more referrals for genetic testing for BRCA1 versus BRCA2 breast cancer patients. 

Our data strongly suggest that the frequency of gBRCA2 carriage and ER+ tumors among women with breast cancer is underestimated because classical genetic testing criteria are often missing in gBRCA2 families. In a previous review of the literature, we found that 50% of overall *BRCA1/2* breast cancer cases are missed when genetic testing criteria are used [32]. Accordingly, the study by Li et al. [25] showed that a higher percentage of BRCA2 versus BRCA1 carriage (81% vs. 46%) was missed by clinical screening. In the study of unselected breast cancer cases by Beitsch et al. [20], which excluded patients who had previously been tested, 86% of the gBRCA mutation cases were gBRCA2. In this study, which enrolled about 1000 patients with breast cancer diagnosis, 50% of women met National Comprehensive Cancer Network (NCCN) criteria for BRCA testing and 50% did not. Overall, 8.6% carried a gBRCA; 9.4% of patients who met NCCN criteria carried a gBRCA, whereas 7.9% of patients who did not meet NCCN criteria carried a gBRCA. Together, these results suggest that 45% of patients with actionable gBRCA mutations, mostly *BRCA2*, are being missed when testing strategies focus on NCCN or other hereditary genetic guidelines. Thus, testing criteria in women with newly diagnosed breast cancer should be expanded. Clear genetic information concerning individual and family consequences of identifying a gBRCA, including preventive end screening strategy and information to siblings, should be given and informed consent obtained from all patients for whom a genetic test is proposed. Specialized genetic counselling should be proposed with a strong family history or if additional information is required. 

This study shows that the frequency of gBRCA2 carriage is higher than expected in both unselected breast cancer patients and in the general population. It also shows that more than half of breast cancer cases occurring in women with a gBRCA are in fact ER+. Because of the important benefit of identifying gBRCA carriage for PARPi treatment and prevention, ER+ breast cancer should be fully considered for BRCA screening.

## Figures and Tables

**Figure 1 cancers-14-03266-f001:**
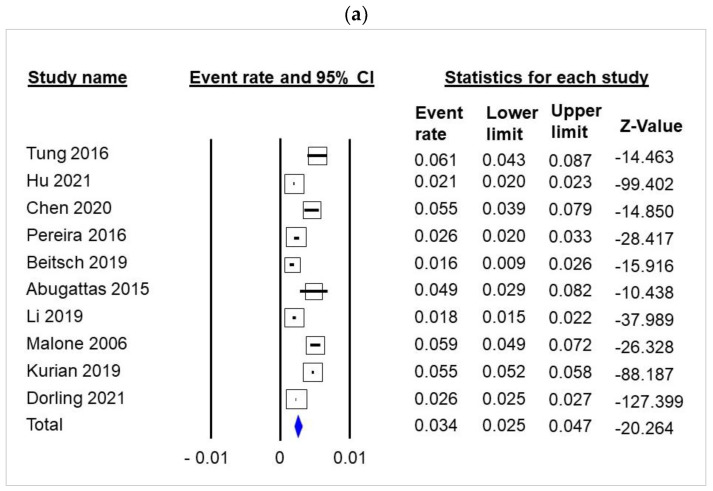
Meta-analysis of gBRCA1 and gBRCA2 among 108,699 unselected breast cancer patients. (**a**) gBRCA; (**b**) gBRCA1; (**c**) gBRCA2.

**Figure 2 cancers-14-03266-f002:**
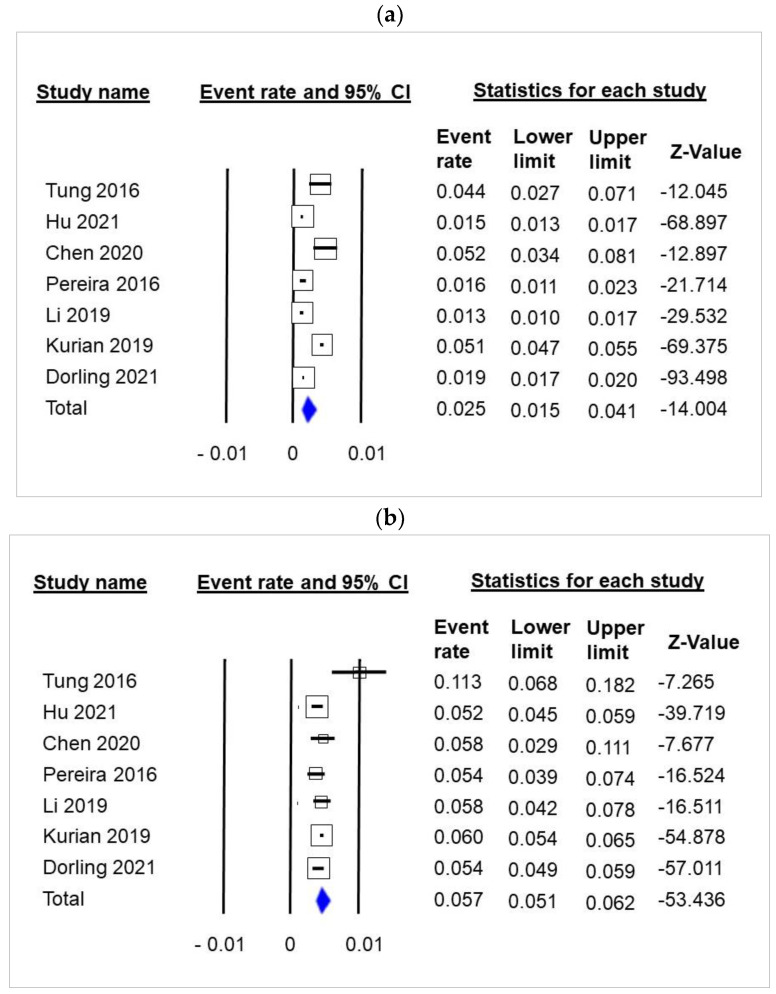
Meta-analysis of gBRCA1 and gBRCA2 among unselected breast cancer patients by estrogen receptor (ER) status. (**a**) gBRCA carriers among ER+ unselected breast cancer patients; (**b**) gBRCA carriers among ER- unselected breast cancer patients; (**c**) Frequency of ER+ and ER- tumors among unselected breast cancer patients with gBRCA. Legend: n of tumors are in bracket. The total number of breast cancer cases analyzed was 86,670.

**Figure 3 cancers-14-03266-f003:**
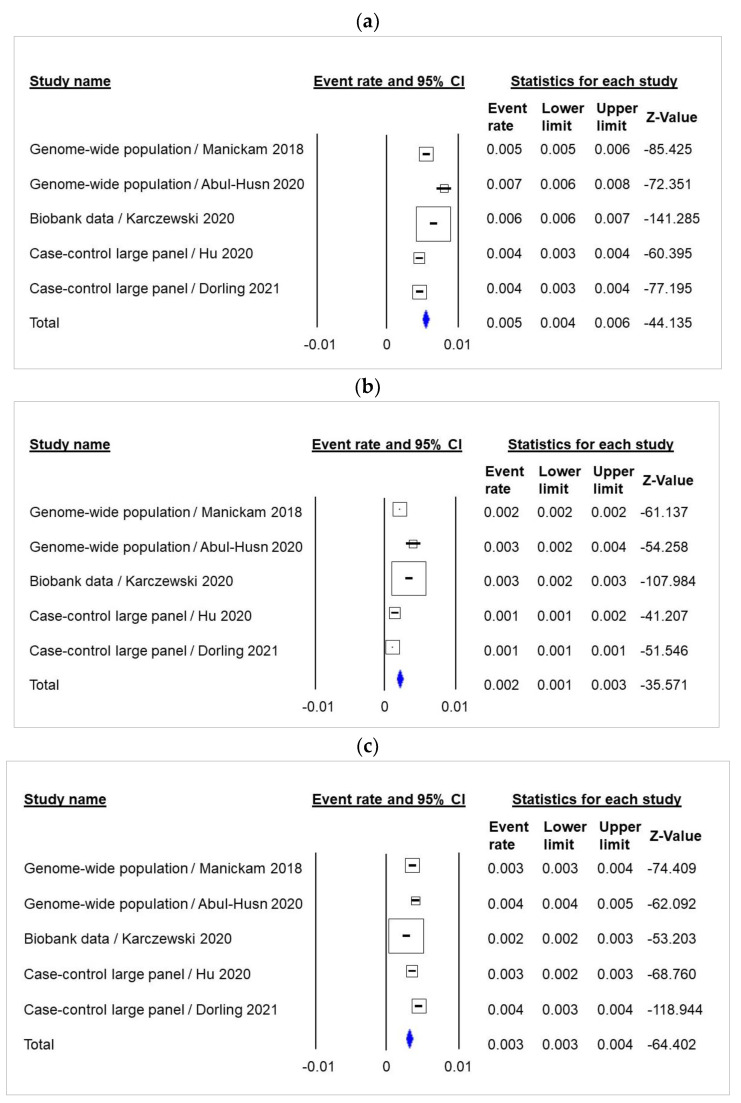
Meta-analysis of gBRCA1 and gBRCA2 carriers among 238,972 unaffected individuals. (**a**) gBRCA carriers among unaffected individuals; (**b**) gBRCA1 carriers among unaffected individuals; (**c**) gBRCA2 carriers among unaffected individuals.

**Figure 4 cancers-14-03266-f004:**
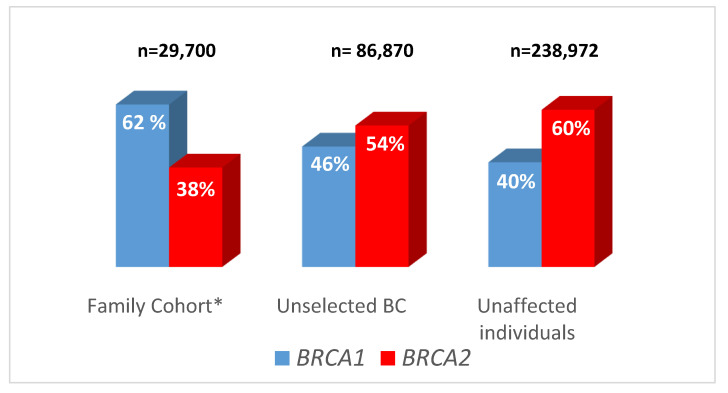
gBRCA1 and gBRCA2 frequency among family cohorts, unselected breast cancer patients and unaffected individuals. * Family cohort: 29,700 gBRCA families from the CIMBA cohort (6).

## Data Availability

Not applicable.

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
