# Peer review of "Predominance of BRCA2 Mutation and Estrogen Receptor Positivity in Unselected Breast Cancer with BRCA1 or BRCA2 Mutation"

_cancers, 2022, doi:10.3390/cancers14133266_

Round 1
Reviewer 1 Report
Pujol et al performed a meta-analysis of published studies and databases to evaluate the relative distribution of gBRCA1 and gBRCA2 in unselected populations of women with breast cancer and in unaffected individuals. They also analyze the proportion of estrogen receptor (ER) positive (ER+) tumors in unselected breast cancer patients with gBRCA. In both cohorts (unselected BC and unaffected individuals) a higher proportion of gBRCA2 was encountered, when compared to familial studies of BRCA carriers. Also, a higher proportion of ER+ patients in gBRCA1 was described, when comparing to selected cohorts. The authors discuss how this imbalance is likely caused by referral bias, since tumors in gBRCA1 carriers are more likely to have triple-negative histology, occur in earlier ages and be associated to family history of ovarian cancer. The overall conclusions of the paper points to the need of expanding testing criteria in women with newly diagnosed breast cancer, a very important issue that is discussed in the context of PARP inhibitors therapy.
Minor corrections:
I believe the title does not capture the overall goal and design of the study. Instead of “among BRCA1/2 mutation carriers”, maybe it should be specified that unselected breast cancer patients were evaluated.
In supplementary table 2, it would be useful to have a description of country of origin or ethnicities of the cohorts of each study.
A paragraph discussing about possible “sporadic” breast cancer patients developed in gBRCA carriers, with tumors displaying absence of second hits in the affected gene and how they would be expected to respond to PARPi would be of interest. Is there evidence of smaller rates of second hits in gBRCA1 ER+ tumors compared to triple-negative ones?
Gene names need to be italicized in several parts of the manuscript.
Page 4, 4th paragraph: define PTC
Page 10, 2nd paragraph: include CIMBA reference.
Author Response
Following are our responses by item
I believe the title does not capture the overall goal and design of the study. Instead of “among BRCA1/2 mutation carriers”, maybe it should be specified that unselected breast cancer patients were evaluated.
We fully agree and thank the reviewer for this comment. The proposed title is now as follows:
Predominance of BRCA2 mutation and estrogen receptor positivity in unselected breast cancer with BRCA1 or BRCA2 mutation.
In supplementary table 2, it would be useful to have a description of country of origin or ethnicities of the cohorts of each study.
A column has been added to table 2 indicating countries of origin the population, and when available, ethnicity.
Modified table 2 :
|
Study |
Country (Ethnicity) |
ER+ * |
BRCA1/ n |
BRCA2/ n |
ER- |
BRCA1/ |
BRCA2/ |
|
Tung 2016 |
United States (Ashkenazi Jewish 7.8%, non-Ashkenazi white 81.4%, other 10.8%) |
364 |
5 |
11 |
124 |
13 |
1 |
|
Hu 2021 |
United States (Asian 4%, Non-Hispanic black 12.3%, Hispanic 3.2%, Non-Hispanic white 78.9%, Other 1.7%) |
18,428 |
73 |
201 |
3805 |
114 |
82 |
|
Chen 2020 |
South China |
363 |
4 |
15 |
139 |
7 |
1 |
|
Li 2019 |
Sweden |
3637 |
17 |
30 |
643 |
30 |
7 |
|
Kurian 2019 |
United States |
11642 |
227 |
365 |
7060 |
309 |
112 |
|
BCAC** 2021 |
16 European countries, 5 Asian countries, 3 American countries 2 Oceania countries |
30466 |
120 |
446 |
7766 |
269 |
149 |
|
TOTAL |
|
66680 |
457 |
1085 |
20190 |
761 |
368 |
* including ER+PR- and ER+HER2+++ tumors; ** BCAC: Breast Cancer Association Consortium
A paragraph discussing about possible “sporadic” breast cancer patients developed in gBRCA carriers, with tumors displaying absence of second hits in the affected gene and how they would be expected to respond to PARPi would be of interest. Is there evidence of smaller rates of second hits in gBRCA1 ER+ tumors compared to triple-negative ones?
A paragraph addressing this important point has been added to the discussion. It is assumed that sporadic cases may have the same frequency in both BRCA1 and BRCA2 carriers. To our knowledge, there are few data on the question of a second hit in terms of ER phenotype. However, it is likely that not all breast tumors display a double hit feature, which suggests that a subset of the BRCA-related breast cancer may develop (and possibly be sensitive to PARPi) in the absence of a BRCA second event (Osorio et al., 2002; Palacios et al., 2003; Tung et al., 2010; Stefansson et al., 2011; Martins et al., 2012). Maxwell et al. (2017) analyzed BRCA germline mutated breast and ovarian tumors and found that although gBRCA1 breast and ovarian tumors had LOH in 90% and 93% of cases, respectively, gBRCA2 tumors retained the wild-type allele in 16% of all gBRCA2 ovarian tumors and 46% of gBRCA2 breast tumors. Thus, the question of a second hit in terms of ER phenotype deserved to be further investigated.
Corrections have been made on the following points:
- Gene names need to be italicized in several parts of the manuscript.
- Page 4, 4thparagraph: define PTC
- Page 10, 2ndparagraph: include CIMBA reference.
Reviewer 2 Report
This manuscript is a purely meta-analysis study to evaluate the status of BRCA1 and BRCA2 germline mutation in unselected populations of women with breast cancer and in unaffected individuals using available data from 31 published paper. The author also conducted analysis on tumor types stratified by ER status.
I found several issues with this manuscript:
· Pure meta-analysis study using data already available. Basically just adding up numbers to get the same conclusions from the papers they pooled. There is no sufficiently new insight gained from this manuscript.
· Inconsistent writing makes it very confusing to read and get the logic. There are several typos in the main text, non-matching hazard ratio (Figure S1. Embraca), and inconsistency of presenting the values. For example, the authors use percentage and fraction to present the frequency of gBRCA1 and gBRCA2.
· Suboptimal figure arrangement. Not aligned titles for subfigures, bad quality, and text size all different. Label inconsistency in Figure 3, apparently 3a, 3b, and 3c used the same set of data but the label format for study name is not the same.
· The quoting format for the study name is also very inconsistent. The authors used first author for some studies and corresponding authors for others. For instance, the author used a publication labeled as “Easton 2021”, which contain one of the largest n among the selected papers used in this meta-analysis. I tried to search “Easton” for the reference and it did not show up in the reference.
· Figure 2c is misleading. The pie chart was draw based on unselected breast cancer patients with gBRCA but the n they shown represent the total breast cancer patients in the meta-analysis.

Author Response
Following are our responses by item
Pure meta-analysis study using data already available. Basically just adding up numbers to get the same conclusions from the papers they pooled. There is no sufficiently new insight gained from this manuscript.
We believe that the principle informations from this meta-analysis are as follow:
- the frequency of gBRCA-related BC among unselected BC is 3.4% (in a population of 108,699 unselected breast cancer)
- the frequency of gBRCA carriers among unaffected individuals is 0.5% (in a population of 238,972 unaffected individuals)
3) gBRCA2 is predominant as compared with gBRCA1 in both unselected BC and unaffected individuals
4) more than half of BRCA1/2-related breast tumors in fact feature an ER+ phenotype
We believe that the results are of clinical interest in the era of the therapeutic value of BRCA PV because the testing for theragnostic purposes (and also according to preventive criteria) may be largely influenced by triple negativity of BC or young age, which are features of BRCA1 BC.
For unaffected individuals, the meta-analysis shows that the heterozygote frequency of gBRCA may have been underestimated, particularly for BRCA2 carriers.
The results of the meta-analysis of more than 100,000 unselected breast cancer cases and 200,000 genomes of unaffected individuals is to our knowledge the largest epidemiological study available.
Inconsistent writing makes it very confusing to read and get the logic. There are several typos in the main text, non-matching hazard ratio (Figure S1. Embraca),
We corrected typos in the main text and we apologize for that.
We checked the hazard ratio of Embraca subgroup in figure S1 but could not find a non-matching hazard ratio. We hope we did not miss something. The subgroup analysis of HR+ shows a hazard ratio of 0.47 (95%CI 0.32-0.71).
Inconsistency of presenting the values. For example, the authors use percentage and fraction to present the frequency of gBRCA1 and gBRCA2.
Values have been homogenized. Frequency of heterozygote are commonly given as fraction. Since the addition of BRCA1 and BRCA2 was exactly equal to 0.5% in unaffected individuals, we indicated this percentage in order to make the represention simpler.
Suboptimal figure arrangement. Not aligned titles for subfigures, bad quality, and text size all different. Label inconsistency in Figure 3, apparently 3a, 3b, and 3c used the same set of data but the label format for study name is not the same.
We deeply apologize for that and thank the reviewer for pointing out this inconsistency of form. All figures have been carefully homogenized throughout the manuscript, including :
Fig1a, 1b, 1c, 2a, 2b
Table S1
Fig S5, S6, S7, S8, S9, S10
The quoting format for the study name is also very inconsistent. The authors used first author for some studies and corresponding authors for others. For instance, the author used a publication labeled as “Easton 2021”, which contain one of the largest n among the selected papers used in this meta-analysis. I tried to search “Easton” for the reference and it did not show up in the reference.
This study mentioned is from the breast cancer association consortium, including more than 200 authors. Therefore, we improperly chose the last and “senior” author. We corrected this in the manuscript and table (now BCAC).
Breast Cancer Association Consortium; Dorling, L.; Carvalho, S.; Allen, J.; González-Neira, A.; Luccarini, C.; Wahlström, C.; Pooley, K.A.; Parsons, M.T.; Fortuno, C.; et al. Breast Cancer Risk Genes - Association Analysis in More than 113,000 Women. N. Engl. J. Med. 2021, 384, 428–439
Figure 2c is misleading. The pie chart was draw based on unselected breast cancer patients with gBRCA but the n they shown represent the total breast cancer patients in the meta-analysis.
We acknowledge that this could be misleading. To clarify, we proposed to put the exact number of patients in BRCA/ER subtypes in the figure (see modified fig 3C, n are in bracket) and the total n of unselected BC with ER status in the legend.
This manuscript is a resubmission of an earlier submission. The following is a list of the peer review reports and author responses from that submission.